# Risk of death following COVID-19 vaccination or positive SARS-CoV-2 test in young people in England

Vahé Nafilyan [1,2,6] ✉, Charlotte R. Bermingham [1,6] ✉, Isobel L. Ward[1], Jasper Morgan [1], Francesco Zaccardi[3], Kamlesh Khunti [3], Julie Stanborough[1], Amitava Banerjee [4,7] & James C. Doidge [2,5,7]

Several studies have reported associations between COVID-19 vaccination and risk of cardiac diseases, especially in young people; the impact on mortality, however, remains unclear. We use national, linked electronic health data in England to assess the impact of COVID-19 vaccination and positive SARS-CoV-2 tests on the risk of cardiac and all-cause mortality in young people (12 to 29 years) using a self-controlled case series design. Here, we show there is no significant increase in cardiac or all-cause mortality in the 12 weeks following COVID-19 vaccination compared to more than 12 weeks after any dose. However, we find an increase in cardiac death in women after a first dose of non mRNA vaccines. A positive SARS-CoV-2 test is associated with increased cardiac and all-cause mortality among people vaccinated or unvaccinated at time of testing.

On the 8 December 2020, the United Kingdom (UK) began administering vaccines against COVID-19 according to the priority groups determined by the Joint Committee on Vaccination and Immunisation (JCVI)[1]. While the randomised clinical trials focused on short-term efficacy against symptomatic infection, which was subsequently found to wane and be escaped by SARS-CoV-2 variants[2–7], real-world evidence has indicated stronger and more sustained effectiveness against severe disease and death due to COVID-19[6,8,9]. However, it is also important to consider their safety, which can be difficult to assess in randomised clinical trials that are not powered to detect rare adverse events[2,3,10].

There have been rare cases of serious adverse events reported with the COVID-19 vaccines. Previous studies have shown an increase in the risk of myocarditis and myopericarditis associated with mRNA vaccines including BNT162b2 (Pfizer-BioNTech) and mRNA-1273 (Moderna)[8,11], and an increased risk of thrombotic and other rare cardiovascular events after the ChAdOx1 nCoV-19 vaccine (Oxford-AstraZeneca)[12,13]. There is also evidence of a range of other rare neurological complications[14]. However, the absolute risk of severe complications is low and needs to be assessed against the increased risks associated with SARS-CoV-2 infection if unvaccinated[15,16]. The balance of risk and benefit is particularly important to determine in younger people, due to the lower risk of COVID-19 hospitalisation and death in this age group[17].

Comparisons of the risk of death in vaccinated and unvaccinated young people are subject to confounding due to the vaccine prioritisation of, and higher vaccination rates among, those with underlying health conditions. To minimise potential confounding, we used a self-controlled case series (SCCS) design, where each participant acts as their own control, to compare the risk of death in the twelve weeks after vaccination ('risk period') to a subsequent reference period[18]. For comparison purposes, we assessed the impact of a positive test for SARS-CoV-2 on the same outcomes in vaccinated and unvaccinated individuals.

[1]Data and Analysis for Social Care and Health, Office for National Statistics, Newport NP10 8XG, UK. [2]Department of Medical Statistics, London School of Hygiene and Tropical Medicine, London, UK. [3]Real World Evidence Unit, Diabetes Research Centre, University of Leicester, Leicester, UK. [4]Institute of Health Informatics, University College London, London NW1 2DA, UK. [5]Intensive Care National Audit and Research Centre, London, UK. [6]These authors contributed equally: Vahé Nafilyan, Charlotte R. Bermingham. [7]These authors jointly supervised this work: Amitava Banerjee, James C. Doidge. ✉ e-mail: vahe.nafilyan@ons.gov.uk; charlotte.bermingham@ons.gov.uk

Here, we show that among the population of England aged 12–29, mortality is not significantly increased in the first twelve weeks after COVID-19 vaccination compared with more than 12 weeks after any dose. Observed reductions in all-cause mortality are consistent with a time-varying healthy vaccinee effect due to postponement of vaccination during periods of poor health. However, in subgroup analyses we find a significant elevation in the risk of cardiac death in women after a first dose of non-mRNA vaccines, and a smaller, non-significant increase in cardiac death after second dose of mRNA vaccines in men. By contrast, a positive SARS-CoV-2 test is associated with increased cardiac and all-cause mortality among both vaccinated and unvaccinated individuals.

## Results

### Characteristics of the study population

There were 3807 deaths of 12 to 29 year-olds in England that occurred between 8 December 2020 and 25 May 2022 and were registered by 8 June 2022 (Supplementary Fig. 1). Of these, 444 (11.7%) were due to a cardiac event and 1512 (39.7%) were linked to a vaccination record (1510 from NIMS and 2 from the supplementary NHS point of care extract) (Table 1). 62.8% (950) of first doses, 51.6% (505) of second doses and 98.8% (239) of third doses in the death registrations dataset were mRNA based (either the BNT162b2 Pfizer-BioNTech or mRNA-1273 Moderna vaccines), rather than non-mRNA based (the ChAdOx1 Oxford-AstraZeneca vaccine) or another vaccine or unknown. Of those who received both the first and second vaccination (979), 11.3% (111) received a different type of vaccine for each dose (Supplementary Table 1). Counts of deaths by sex and vaccine vector for people who received at least one dose of that vector are included (Supplementary Table 2).

Between 8 December 2020 and 31 March 2022, 1420 hospital deaths among 12- to 29-year-olds were recorded in HES, 630 (44.4%) of which were linked to a vaccination record (629 from NIMS and 1 from the supplementary NHS point of care extract). 63.3% (399) of first doses, 53.5% (228) of second doses and 98.2% (108) of third doses in the hospital deaths dataset were mRNA based. Of those who received both the first and second vaccination (422), 12.0% (51) received a different type of vaccine for each dose (Supplementary Table 1). Participant characteristics were similar across the three case-series (Table 1).

The corresponding analyses of deaths after positive SARS-CoV-2 test, between 8 December 2020 and 31 December 2021, included 3219 registered deaths, of which 369 (11.5%) were due to a cardiac event, and

353 (11.0%) were linked to a preceding positive SARS-CoV-2 test (297 (9.2%) occurred unvaccinated at the date of test registration and 56 (1.7%) vaccinated at the date of test registration). There were 1123 corresponding hospital deaths recorded in HES between 8 December 2020 and 31 December 2021 of individuals who were not infected on admission to hospital, of which 181 (16.1%) were linked to a preceding positive SARS-CoV-2 test (150 (13.4%) unvaccinated on date of test registration and 31 (2.8%) vaccinated). Participant characteristics were again similar across the three case series (Table 2).

Consistent with an increasing prevalence of registration delay due to coroner referral associated with more recent deaths, the observed incidence of ONS registered deaths decreased over time (Supplementary Fig. 2). Hospital deaths recorded in HES are unaffected by coroner delays and did not exhibit such a trend. Similar patterns were observed for the analyses of deaths after a positive SARS-CoV-2 test for the registered deaths and hospital deaths datasets (Supplementary Fig. 3).

Supplementary Fig. 4A shows the number of deaths each week since vaccination. There was a marked decline at around 12 weeks after first dose, when most people would have a second dose. Supplementary Fig. 4B shows the number of deaths each week since positive SARS-CoV-2 test, with markedly higher numbers of all-cause deaths occurring in the first 5 weeks.

### Relative incidence of death after COVID-19 vaccination

In the twelve weeks post-vaccination compared with subsequent periods, there were no significant increases in the incidence of any mortality outcome for all vaccine doses combined across each individual week or all twelve weeks combined (all-cause registered death: incidence rate ratio, IRR, 0.88, 95% confidence interval, CI, 0.80, 0.97; cardiac registered death: IRR 1.11, 95% CI 0.87, 1.42; all-cause hospital death: IRR 0.89, 95% CI 0.77, 1.04) (Fig. 1a). There were also no significant increases in incidence of any mortality outcome for any individual dose for all twelve weeks combined or across each individual week in the first 12 weeks after vaccination, except week 12 for cardiac deaths after dose 1 (IRR 3.22 95% CI 1.39, 7.48; Fig. 1b).

We found a significant decrease in the incidence of all-cause registered death, driven by the first two weeks after vaccination (any dose, week 1: IRR 0.47 [0.34, 0.64]; week 2: 0.77 [0.60, 0.99]; Fig. 1a). Similarly, there was a decreased risk of hospital death in the first two weeks after vaccination (any dose, week 1: IRR 0.32 [0.18, 0.60]; week 2: 0.51 [0.31, 0.83]; Fig. 1a).

**Table 1 | Characteristics of the study population for deaths after vaccination, young people aged 12-29, resident in England, who died between 8 December 2020 – 25 May 2022 (registered by 8 June 2022) for ONS registered deaths, and who died 8 December 2020 – 31 March 2022 for HES hospital deaths**

| Characteristic | Level | All-cause registered deaths | Cardiac registered deaths | All-cause hospital deaths |
|---|---|---|---|---|
| Total | | 3807 | 444 | 1420 |
| Age group | 12–17 | 544 (14.29) | 58 (13.06) | 233 (16.41) |
| | 18–24 | 1548 (40.66) | 161 (36.26) | 566 (39.86) |
| | 25–29 | 1715 (45.05) | 225 (50.68) | 621 (43.73) |
| Risk period | Unvaccinated | 2295 (60.28) | 234 (52.70) | 790 (55.63) |
| | 12 weeks or less | 791 (20.78) | 123 (27.70) | 323 (22.75) |
| | 13+ weeks | 721 (18.94) | 87 (19.59) | 307 (21.62) |
| Sex | Male | 2524 (66.30) | 296 (66.67) | 841 (59.23) |
| | Female | 1283 (33.70) | 148 (33.33) | 579 (40.77) |
| Most recent vaccination | First | 533 (14.00) | 62 (13.96) | 204 (14.37) |
| | Second | 737 (16.36) | 107 (24.10) | 316 (22.25) |
| | Third | 242 (6.36) | 41 (9.23) | 110 (7.75) |
| | Unvaccinated | 2295 (60.28) | 234 (52.70) | 790 (55.63) |

Figures are number (%) unless stated otherwise.

**Table 2 | Characteristics of the study population for deaths after SARS-Cov-2 positive test, young people aged 12-29, resident in England, who died between 8 December 2020 - 31 December 2021 for ONS registered deaths and HES hospital deaths. Figures are number (%) unless stated otherwise**

| Characteristic | Level | All-cause registered deaths | Cardiac registered deaths | All-cause hospital deaths |
|---|---|---|---|---|
| Total | | 3219 | 369 | 1123 |
| Age group | 12–17 | 427 (13.26) | 46 (12.47) | 175 (15.58) |
| | 18–24 | 1333 (41.41) | 130 (35.23) | 449 (39.98) |
| | 25–29 | 1459 (45.32) | 193 (52.30) | 499 (44.43) |
| Positive SARS-CoV-2 test | No positive test | 2866 (89.03) | 328 (88.89) | 942 (83.88) |
| | Positive test (unvaccinated at infection) | 297 (9.23) | – | 150 (13.36) |
| | Positive test (vaccinated at infection) | 56 (1.74) | – | 31 (2.76) |
| Risk period | No positive test | 2886 (89.03) | 328 (88.89) | 942 (83.88) |
| | 12 weeks or less (unvaccinated at infection) | 158 (4.91) | – | 96 (8.55) |
| | 12 weeks or less (vaccinated at infection) | 41 (1.27) | – | 23 (2.05) |
| | 13+ weeks (unvaccinated at infection) | 139 (4.32) | – | 54 (4.81) |
| | 13+ weeks (vaccinated at infection) | 15 (0.47) | – | 8 (0.71) |
| Sex | Male | 2161 (67.13) | 240 (65.04) | 673 (59.93) |
| | Female | 1058 (32.87) | 129 (34.96) | 450 (40.07) |

Some values are supressed due to low counts.

Subgroup analyses by vaccine vector were generally consistent with the main results. However, for non-mRNA or unknown vaccine vectors (primarily consisting of the ChAdOx1 Oxford-AstraZeneca) for the first dose and all doses combined, there were significant increases in the incidence of cardiac death (first dose: IRR 1.75 [1.14, 2.71]; all doses: 1.71 [1.20, 2.45]; Fig. 2a). A raised risk of all-cause hospital death was observed after a first dose of a non mRNA vaccine (IRR 1.33 [1.01, 1.74]), but this was not seen in all-cause registered deaths (IRR 0.90 [0.75, 1.07]). Subgroup analyses stratified by age group and sex were consistent with the main results except after the first dose in females, in which cardiac death was increased (IRR 1.79 [1.05, 3.05]). This corresponded to 1 additional cardiac registered death for every 363,419 (95% CI 238,784, 3,272,867) females aged 12–29 vaccinated (Fig. 2b). However, there was no evidence of a raised risk in females after second or third doses (IRR 0.73 [0.37,1.43] and 1.28 [0.32, 5.17] respectively) or when looking at all doses combined (IRR 1.31 [0.86, 2.00]). Additionally, there was no evidence of elevated all-cause mortality in this subgroup after the first dose (all-cause registered death: IRR 1.07 [0.88, 1.31]; all-cause hospital death: IRR 1.01 [0.74, 1.38]).

Subgroup analyses by vaccine vector broken down by sex (Supplementary Fig. 5) showed a raised risk of cardiac death in females following a first dose of a non mRNA vaccine and for all doses combined (first dose: IRR 3.52 [1.71 – 7.26]; all doses: 3.02 [1.65, 5.53]) but not after a first dose or any dose of mRNA vaccine (first dose: IRR 0.87 [0.41 – 1.85]; all doses: 0.76 [0.43, 1.34]). However, no significant increase in the risk of cardiac death was observed in females after the second dose of an mRNA vaccine (IRR 0.59 [0.24, 1.44]) and a non-significant increase was observed after a first dose of a non-mRNA vaccine (1.88 (0.73, 4.87)). This increase in risk corresponded to 1 additional cardiac registered death for every 16,486 (95% CI 13,688, 28,426) females aged 12–29 who received a first dose of a non mRNA vaccine. An increased risk of all-cause hospital death was also observed in females after a first dose, and after any dose of a non-mRNA vaccine (first dose: IRR 1.66 [1.10 – 2.51]; any dose: 1.55 [1.10, 2.18]), however this was not observed in all-cause registered deaths (first dose: IRR 1.13 [0.85, 1.49]; any dose: 1.09 [0.86, 1.37]).

There was also a non-significant elevation in IRR for cardiac death following a second dose of an mRNA vaccine in men (IRR 1.70 [0.98, 2.97]), which, if valid, would correspond to one additional cardiac registered death for every 359,294 (223,043, -) men aged 12-29 who received a second dose of an mRNA vaccine. This result was sensitive

to variation of the risk period, with no increase observed with a 6 week risk period (IRR 1.15 (0.60, 2.18)). (Supplementary Fig. 6).

**Relative incidence of death after positive test for SARS-CoV-2**
Following a positive SARS-CoV-2 test among individuals unvaccinated at date of test registration, there was an increase in the incidence of cardiac death (IRR for the whole 12-week period: IRR 2.35 [1.09, 5.06]), driven by the first week (IRR 11.56 [3.93, 33.99]; Fig. 3a). During the whole 12-week period, higher incidences of all-cause mortality were also observed for both registered deaths (IRR 2.50 [1.93, 3.23] and hospital deaths (IRR 4.50 [3.09, 6.54]), similarly most pronounced in the first week (IRR 6.87 [4.53, 10.42] and 9.02 [4.79, 16.97], respectively).

There was also an increase in the incidence of all-cause death following a positive SARS-CoV-2-test among individuals vaccinated at date of test registration (IRR 1.94 [1.03, 3.67] for all-cause registered deaths; 2.76 [1.14, 6.71] for all-cause hospital deaths; Fig. 3b), similarly most pronounced in the first two weeks. There were insufficient data to analyse cardiac deaths among people vaccinated on the date of test registration.

Relative increases in mortality rates following a positive SARS-CoV-2 test were comparable across all subgroups but with greater uncertainty in younger age groups (Fig. 4).

In people aged 12-29 with a positive SARS-CoV-2 test, the increased risk of all-cause registered death in the following twelve weeks corresponded to 1 additional death for every 11,936 (95% CI 10,373, 14,862) individuals aged 12–29 and unvaccinated on date of positive test registration, and 1 additional death for every 55,661 (37,071, 925,962) individuals aged 12–29 and vaccinated on date of positive test registration.

**Sensitivity analyses**
For all three analyses of death after vaccination, varying defined lengths of risk period were consistent with the main findings in remaining below or not significantly different to 1 (Supplementary Fig. 7). For analyses of deaths after a positive SARS-CoV-2 test for individuals unvaccinated at the date of test registration, estimates of relative incidence of all-cause death were greater for shorter risk period definitions, particularly for hospital deaths. While convergence towards one was observed as the defined risk period was increased, complete convergence was not observed, with the relative incidence

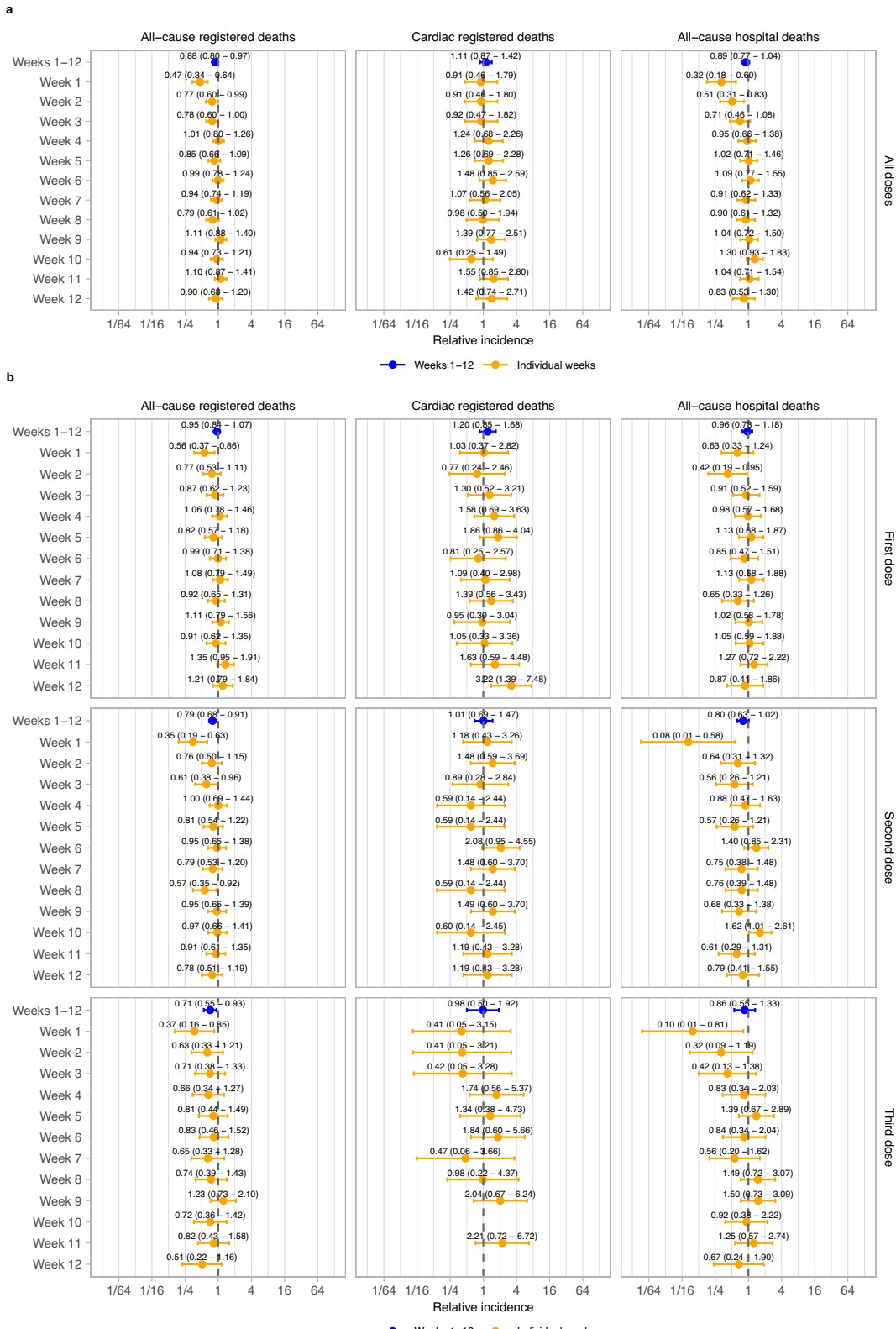

**Fig. 1 | Relative incidence of cardiac death and all-cause death in each of the 12 weeks in the risk period after vaccination and in the risk period as a whole (1-12 weeks), compared to the baseline period.** Data are presented as incidence rate ratio with 95% confidence intervals. Models are adjusted for calendar time; IRR not shown if no event. *n* = 3807 all-cause registered deaths, 444 cardiac registered deaths and 1420 all-cause hospital deaths. **a** Results for all doses combined. **b** Results by dose.

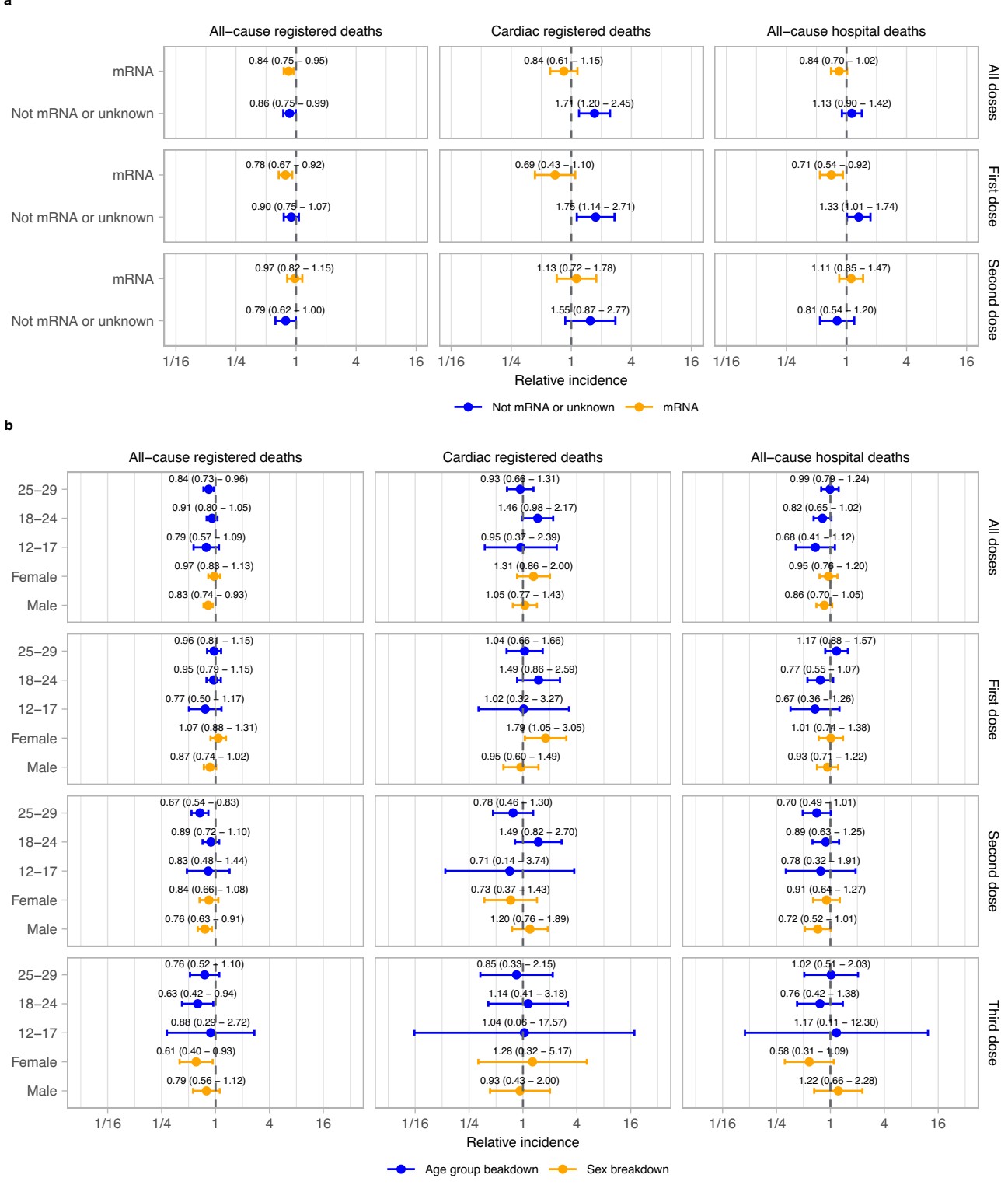

**Fig. 2 | Relative incidence of cardiac and all-cause death in the risk period after vaccination, compared to the baseline period, by sex, age group and vaccine vector.** Data are presented as incidence rate ratio with 95% confidence intervals. Models are adjusted for calendar time. *n* = 3807 all-cause registered deaths, 444 cardiac registered deaths and 1420 all-cause hospital deaths. **a** Breakdowns by vaccine vector **b** Breakdowns by sex and age-group. Results are not presented for the third dose because these are almost all mRNA based.

appearing to stabilise above one for all-cause and cardiac deaths (Supplementary Fig. 7A). For analyses of deaths after a positive SARS-CoV-2 test for individuals vaccinated at date of test registration (Supplementary Fig. 8B), estimates of the relative incidence of all-cause registered the death and all-cause hospital death were greater for

shorter risk period definitions and converged to 1 as the risk period was increased.

Results for the analyses of vaccination and positive SARS-CoV-2 tests were consistent when adjusting calendar time in fortnightly intervals compared to the restricted cubic spline adjustment of the

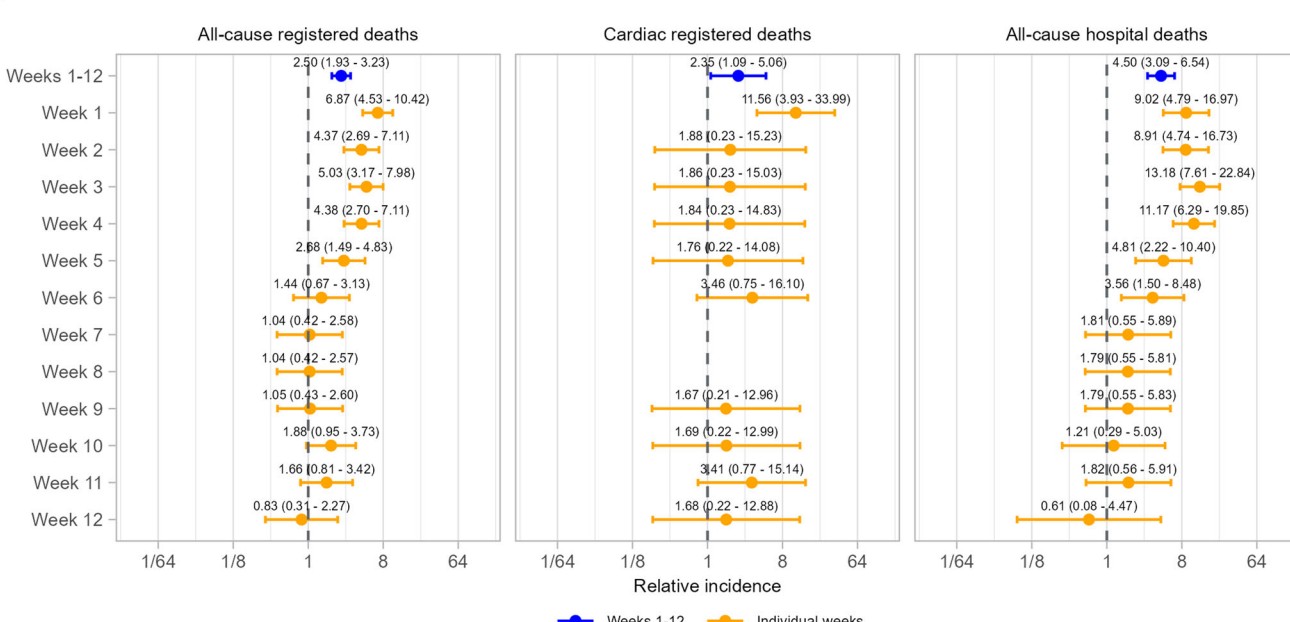

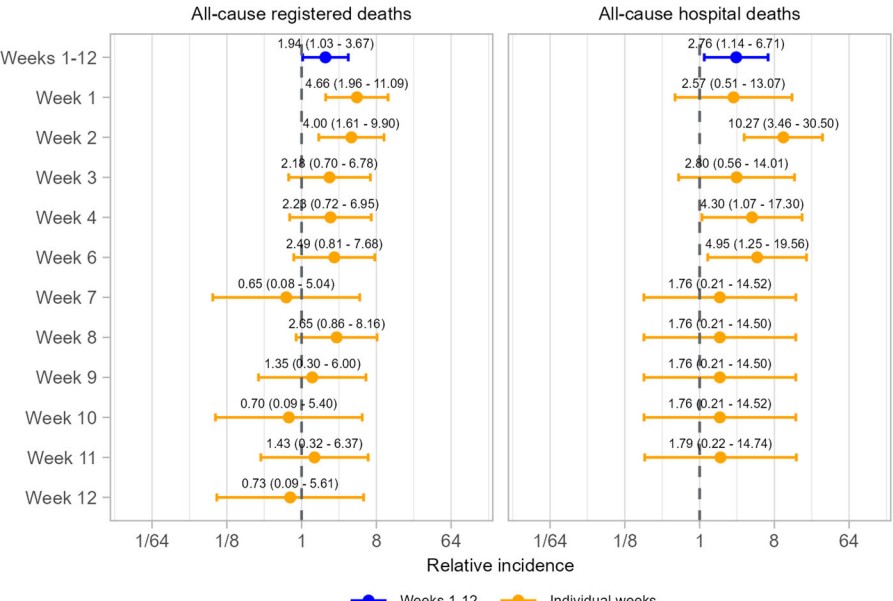

**Fig. 3 | Relative incidence of cardiac death and all-cause death in each of the 12 weeks in the risk period after SARS-CoV-2 infection and in the risk period as a whole, compared to the baseline period.** Data are presented as incidence rate ratio with 95% confidence intervals. Models are adjusted for calendar time. $n$ = 3219 all-cause registered deaths, 369 cardiac registered deaths and 1123 all-cause hospital deaths. **a** Results for individuals unvaccinated at time of SARS-CoV-2 positive test. **b** Results for individuals vaccinated at time of SARS-CoV-2 positive test. Cardiac-registered deaths not shown due to small numbers.

calendar week in the main analysis (Supplementary Figs. 9 and 10). Exclusion of the 23 individuals in the HES analysis of vaccinations who tested positive on the day of hospital admission prior to death gave consistent estimates to the main analysis (Supplementary Fig. 11). Inclusion of the 20 individuals in the HES analysis of positive SARS-CoV-2 tests who tested positive on the day of hospital admission prior to death gave similar but slightly higher estimates for weeks 1-12 compared to the main analysis, but with particularly higher estimates in the first week after testing positive (unvaccinated at date of positive test, weeks 1-12: IRR 5.01 [3.48, 7.23], week 1: IRR 15.26 [8.89, 25.95]; vaccinated at date of positive test, weeks 1–12 IRR: 2.97 [1.24, 7.12]; week 1: IRR 3.90 [0.96, 15.86]) (Supplementary Fig. 12).

## Discussion

Using a self-controlled case series analysis in two independent sources of mortality data, we found minimal evidence of an increased incidence of cardiac or all-cause mortality overall in the twelve weeks following COVID-19 vaccination for all vaccine vectors combined. However, we found evidence of an increase in the risk of all-cause and cardiac death after a first dose of a non-mRNA-based vaccine among females, and some evidence of a smaller increase after a second dose of an mRNA vaccine in males. The increase after the first dose was not observed in the male subgroup for either vaccine vector. The subgroup who received non-mRNA vaccines are more likely to be clinically vulnerable. The ChAdOx1 Oxford Astra-Zeneca vaccine was withdrawn

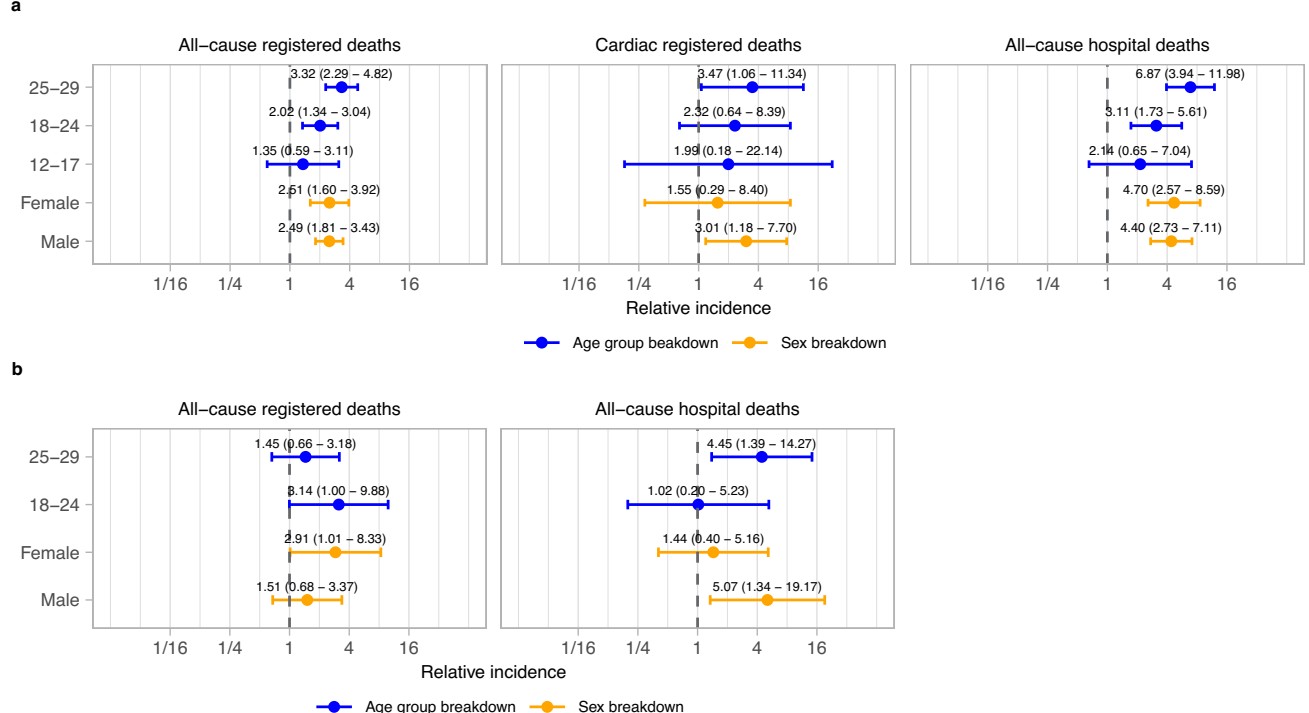

**Fig. 4 | Relative incidence of cardiac and all-cause death in the risk period after SARS-Cov-2 positive test, compared to the baseline period, by sex and age group.** Data are presented as incidence rate ratio with 95% confidence intervals. Models are adjusted for calendar time. *n* = 3219 all-cause registered deaths, 369 cardiac registered deaths and 1123 all-cause hospital deaths. **a** Results for individuals unvaccinated at time of SARS-CoV-2 positive test. **b** Results for individuals vaccinated at time of SARS-CoV-2 positive test. Cardiac-registered deaths not shown due to small numbers.

for people aged under 30 on 7 April 2021[19], and until 8 June 2021 vaccination in this age group was limited to health care workers and people who were clinically extremely vulnerable. People who were clinically extremely vulnerable may be at greater risk of adverse events following vaccination than the general population. In our study population, 52% of people who received a first vaccine before 7 April 2021 received a non mRNA or unknown vaccine compared to 4.3% of people who received a first vaccine on or after 7 April 2021 (Supplementary Table 3). By contrast, we observed an increase in the risk of cardiac and all-cause mortality after a positive SARS-CoV-2 test among both unvaccinated and vaccinated individuals. Results were robust to extending the length of the risk period.

A recent analysis of data from Florida (United States) found increased risk of cardiac death in the first four weeks after mRNA vaccination in people aged 18–39 years[20] compared to subsequent weeks. The Florida study used a SCCS approach but did not account for multiple exposure. Instead, they combined deaths after first and second doses while extending follow-up beyond the interval between doses, which introduced bias. Similarly, excluding patients based on any post-vaccination event (a booster dose or positive test for SARS-CoV-2) would have also biased the vaccination results. A further distinction between the US-based study and our current publication, is the vaccination type; in our study, mRNA vaccination primarily represents the BNT162b2 Pfizer-BioNTech vaccine and non-mRNA vaccination the ChAdOx1 Oxford-AstraZeneca vaccine, whereas in the US, mRNA vaccination represents a larger proportion of the mRNA-1273 Moderna vaccine and non-mRNA primarily represents the Ad26.COV2.S Johnson & Johnson vaccine, and that dosing intervals were shorter in the USA compared to in the UK.

Several other studies have highlighted the association between COVID-19 vaccination and the risk of myocarditis and other cardiac events. Vaccination with mRNA vaccines is associated with an increased risk of myocarditis or myopericarditis, especially in young

people, in investigation using data from the US[21,22], Denmark[11], and England[23], with higher increased risk generally found among young males[24]. We found little evidence of increased risk of death due to cardiac events after an mRNA vaccine for either sex. Whilst there was some indication of a potential increase in risk for men after a second dose of a mRNA vaccine, the result was sensitive to the choice of the risk period.

While vaccinees often have higher rates of comorbidity (and so higher mortality) compared with those who are unvaccinated, early negative effect estimates are frequently observed for safety outcomes in studies estimating vaccine effectiveness—a pattern that is consistent with a time-varying healthy vaccinee effect due to postponement of vaccination during periods of illness or recovery[25].

Our findings of elevated cardiac mortality following positive tests for SARS-CoV-2 are also consistent with evidence from Sweden, showing elevated risk of myocardial infarction and ischaemic stroke following COVID-19;[16] however, ours are based on one of the largest studies of mortality in such a young cohort, where deaths are extremely rare.

While the risk of mortality after positive SARS-CoV-2 tests was most pronounced in the first few weeks, it was not clear from our results whether the excess risk had fully resolved by 12 weeks. A recent matched cohort analysis in an older but geographically similar population, found that the excess risk of cardiovascular disease resolved at around 12 weeks after positive SARS-CoV-2 test[26].

Our study has several strengths. First, we used death registration records and deaths that had occurred in hospital for the whole of England, linked to all vaccination records, including those which were not registered in the NIMS vaccination database because people died shortly after vaccination. Using a self-controlled case series, our estimates account for between-person differences, which is crucial because young people who were clinically extremely vulnerable were prioritised for vaccination.

The main limitation in our analysis of death registration data is delay due to coroner referrals. Not all deaths that occurred in the period had been registered. Deaths of young people are more likely to be referred to the coroner and registration delays can be substantial. Adjustment for calendar time can account for the fact that more recent deaths are more likely to be under coroner investigation and therefore less likely to have been registered. But this adjustment cannot account for any increase in coroner referral related to time since vaccination – for instance, if deaths occurred very soon after vaccination they are more likely to be investigated by a coroner than deaths occurring at any other time. Our analysis using deaths recorded in hospital records, which are not affected by registration delays, was consistent with the analysis of registered deaths, suggesting that differential coroner referral in the period immediately postvaccination did not substantially affect analysis of registered deaths. However, a further limitation of the HES analysis is that sudden cardiac deaths, which would have occurred out of hospital, would not be captured.

Bias could also have been introduced in the HES analyses due to routine testing for SARS-CoV-2 on admission to hospital[27]. Sensitivity tests showed that while this does not significantly affect the results in the main analysis for deaths after vaccination, the estimates for death after a positive SARS-CoV-2 test are inflated, particularly in the first week, due to this bias. Therefore, individuals whose positive SARS-CoV-2 test was on admission to hospital were excluded from the hospital dataset for the analysis of positive tests for SARS-CoV-2.

The SCCS design examines variation in risk over time within individuals and so correct specification of the risk period is crucial. Our sensitivity analyses exploring varying lengths of risk period were all consistent with the main analyses but a very long-lasting increase in risk postvaccination would not have been detectable with this SCCS approach. Conversely, it is also possible that the time-limited healthy vaccinee effect masks a smaller, short-term increase in risk due to vaccination; myocarditis tends to appear very soon after vaccination, with evidence of the median time from vaccination to symptom onset of 2 days[21].

Another implication of the SCCS design is that comparisons cannot be made between the different case series, such as those vaccinated and unvaccinated at time of positive SARS-CoV-2 test. These groups can be expected to differ in important ways such as comorbidity and strain of SARS-CoV-2. Importantly, the difference cannot be interpreted as an estimate of vaccine efficacy.

Lastly, our ability to analyse longer risk periods was limited by the spacing between doses. The negative effects observed in sensitivity analysis with risk periods of more than twelve weeks suggest a negative bias with risk periods that exceeded dosing intervals.

Whilst COVID-19 vaccination has been linked to an increased risk of myocarditis and other cardiac events in young people, we found no evidence of substantially increased mortality risk, either due to cardiac events or overall, from mRNA vaccines, which suggest that cases of myocarditis or myopericarditis due to mRNA COVID-19 vaccines are unlikely to be fatal. We do, however, find evidence of an increased risk of cardiac death after a first dose of a non mRNA vaccine among females. It should also be noted that non mRNA vaccines are no longer used in the UK vaccination programme[28]. This provides reassurance that mRNA vaccines pose minimal risk of increased mortality in the first twelve weeks post-vaccination in young individuals. However, it is important to continue to monitor mortality after vaccination as more deaths are being registered, and extend the surveillance to other age groups and deaths from other causes.

## Methods

Ethical approval was obtained from the National Statistician's Data Ethics Advisory Committee (NSDEC(20)12).

This analysis improves and extends a previous analysis published by the Office for National Statistics[29], by applying a more appropriate SCCS model to an updated outcome dataset, an additional, independent outcome dataset, and additional sensitivity analyses, as described further below.

### Data sources

We separately analysed two independent sources of data on death: deaths registered with the Office for National Statistics (ONS)[30] and hospital deaths recorded in Hospital Episode Statistics for England (HES)[31]. Unlike death registrations, deaths recorded in HES are not subject to delay due to coroner referrals. The deaths datasets included deaths registered by 8 June 2022 that occurred between 8 December 2020 (the start of the vaccine rollout in the England) and 25 May 2022; and HES records where the discharge status indicated deaths between 8 December 2020 and 31 March 2022. Where the same death was recorded in both the registration and hospital datasets, the date of death, date of birth, and sex were taken from the death registration, otherwise these were taken from HES. There were 367 records of deaths in HES (25.8% of the 1420 total) that did not have a linked death registration, likely due to an ongoing coroner inquest.

Death data were linked to data on COVID-19 vaccination from the National Immunisation Management Service (NIMS)[32] and a supplementary extract from NHS point of care data provided by NHS-Digital. The NIMS data includes most COVID-19 vaccinations administered in England since 8 December 2020. However, in rare cases, if the death was recorded on the Personal Demographics Service (PDS) before the vaccination record was sent to NIMS then the patient's vaccination registration in NIMS is not updated. The supplementary extract of NHS point-of-care data includes all vaccination records affected so. The NIMS and point-of-care extracts include all vaccinations recorded by 14 June 2022.

To assess the relative incidence of death following a positive test SARS-CoV-2, we linked death records to national testing data from pillar 1 (tests in hospitals) and pillar 2 (tests in the community)[33] recorded between 9 September 2020 (when mass testing became available) and 31 December 2021. The latter include both laboratory polymerase chain reaction and self-reported rapid antigen tests.

The linkage across databases was conducted using NHS number, which was available for 99.96% of NIMS records, 99.7% of ONS death registrations, 99.1% of deaths in HES, and 100% of the point-of-care extract from NHS-Digital.

Data linkage and preparation were conducted using R 3.5, Python version 3.6 and Spark version 2.4.

### Study population

The study population included all people whose deaths were recorded since the start of the vaccination roll-out on 8 December 2020 and who were aged 12–29 years on the date of death. There were insufficient numbers and follow-up to study vaccination in children below 12 years of age. For the analysis of deaths after a positive SARS-CoV-2 test, we further restricted the study population to deaths that occurred up to 31 December 2021 (the last date on which a positive test could be recorded in our dataset) and individuals whose positive SARS-CoV-2 test was on the day of admission to hospital were excluded. Since the Omicron-dominant period is defined as starting on the 20 December 2021, the data covers mainly the Delta dominant (17 May to 19 December 2021) and Pre-delta (pre-16 May 2021) periods[34]. We separated the analysis for people who were vaccinated or unvaccinated on the day the positive SARS-CoV-2 test was recorded, where vaccinated is having received at least one dose any day before the day of infection.

### Exposure and outcomes

The main exposure was any dose of COVID-19 vaccination within the previous twelve weeks. The comparative exposure was a SARS-CoV-2 positive test within the previous twelve weeks. The date of positive

test is defined as the start of the last episode of COVID-19 for each individual, where a new episode was defined as a positive result more than 120 days after the start of any previous episode. This is in line with the time period for reinfection used in the COVID-19 Infection Survey[35]. Episodes were counted from the first positive test but only those occurring on or after 9 September 2020 were included in the analysis.

Three outcomes were analysed: all-cause registered death and cardiac registered death (ICD-10 code I30-I52 mentioned on the death certificate), from ONS death registrations; and all-cause hospital death, recorded in HES. Each analysis included all participants who experienced the outcome of interest (cases).

## Statistical analysis

We used a SCCS approach designed to handle multiple event-dependent exposures[36]. Event-dependent exposure occurs when the event of interest influences the likelihood of exposure; death represents an extreme example in which any subsequent exposure is impossible after the event. This specific SCCS approach compares the risk of death during a predefined risk period following exposure (vaccination or positive test for SARS-CoV-2) with a reference period of all time subsequent to the risk period. By comparing time periods within individuals, time-invariant factors such as sex and, in general, age group and comorbidity, are implicitly controlled for; time-varying factors can be controlled by the inclusion of unexposed cases in the analysis and by the inclusion of calendar time as a covariate[37].

Participants were followed from 8 December 2020 to the 25 May 2022 for death registrations and 31 March 2022 for hospital deaths and were not censored at death. For the analyses of positive SARS-CoV-2 test, follow up ended at the 31 December 2021 when testing data ended. The risk period was defined as the first twelve weeks after vaccination/positive test while the reference period as all the following weeks to the end of study. In addition, all the weeks prior to vaccination/positive test from 8 December 2020, or all weeks throughout the follow up time if the individual was unvaccinated/never positive, were also included in order to adjust for calendar time.

The exposure was the week since vaccination/positive SARS-CoV-2 test for each of the first twelve weeks individually, or the twelve weeks together. For vaccination, the risk weeks were further categorised by dose or considered for all doses combined.

The SCCS models were fitted using a conditional Poisson regression model using a pseudo-likelihood method on a person-week level dataset[36–39] The calendar day of the start of each week was included in the model using a restricted cubic spline. Adjusting for calendar time is a way to capture the impact of increasing registration delays over time (more recent deaths being more likely to be under coroner review), as well as of seasonal mortality trends and changing SARS-CoV-2 infection rates. The length of each week in days was included as an offset in the model, as some weeks are not complete (such as if a vaccination occurs part way through a week). Incidence rate ratios (IRRs) for cardiac and all-cause deaths in risk periods relative to reference periods were estimated using each model. 95% confidence intervals were obtained from the model estimates. For the main estimates, we also calculated 95% confidence using bootstrapping. Similar analyses were conducted focusing on unvaccinated and vaccinated individuals for deaths after positive test for SARS-CoV-2.

Where the IRR was significantly different from one, we derived estimates of absolute effects using an established method for the self-controlled case series[40]. We used the total number of people who tested positive in pillar 1 and pillar 2 stratified by vaccination status on date of test registration over the period of interest as a measure of exposed cases (Supplementary Methods 1).

All analyses were conducted using R 3.5.

## Subgroup and sensitivity analyses

Analyses were stratified by sex and age group (12-17, 18-24, 25-29). Analyses were also stratified by vaccine vector (mRNA or 'not mRNA or unknown') for the first two doses. 'mRNA' includes the Pfizer BNT162b2 Pfizer-BioNTech and mRNA-1273 Moderna vaccines. 'Not mRNA or unknown' includes the ChAdOx1 Oxford-AstraZeneca vaccine, all other recorded vaccine manufacturers and non-recorded vaccination manufacturers. The analysis of each vaccine vector included all unvaccinated people and all people who received at least one dose of the vaccine vector of interest. The doses were renumbered to correspond to the first, second and third doses of the vaccine vector of interest only. We also break down the analysis of vaccine vector by sex.

We assessed sensitivity to specification of the risk period by exploring a range of different lengths of risk period and to the type of calendar adjustment by adjusting in fortnightly intervals rather than using a restricted cubic spline[36]. We also assessed whether results could be biased by routine testing for SARS-CoV-2 infection in admission to hospital by omitting individuals who had positive test date on the day of admission to hospital for the hospital episode prior to death in the analyses of hospital deaths.

## Patient and public involvement

No patient or member of the public was involved in this study.

## Reporting summary

Further information on research design is available in the Nature Portfolio Reporting Summary linked to this article.

## Data availability

The source data used in this study is subject to controlled access due to its sensitive nature. The ONS is working to make death registration data linked to vaccination data from the National Immunisation Management Service available on the ONS Secure Research Service (SRS). Test and Trace data (unlinked) is also available through the SRS but is not currently linked to deaths registration data in the SRS. Access to the SRS is available to accredited researchers. Details of the application requirements and process, and the use of data, are available at ons.gov.uk/aboutus/whatwedo/statistics/requestingstatistics/securerresearchservice. Microdata on death and vaccination may also be accessed through the NHS-Digital Data Access Request Service. Hospital Episode Statistics data is not available through the ONS; this data is held by NHS England. All statistical data used in this study are available from the Office for National Statistics website.

## Code availability

Code used in this study is available on Github[41].

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

## Acknowledgements

We thank Chris Robertson for useful discussions. K.K. is supported by
the National Institute for Health Research (NIHR) Applied Research
Collaboration East Midlands (ARC EM) and the NIHR Leicester Biomedi-
cal Research Centre (BRC).

## Author contributions

V.N., C.B. and J.D. conceptualised and designed the study. C.B. and J.M.
prepared the study data. C.B. and V.N. performed the statistical analysis,
which were quality checked by I.W. V.N., C.B., I.W., J.M., F.Z., K.K., J.S.,
A.B. and J.C. contributed to interpretation of the findings. V.N. and C.B.
wrote the original draft. V.N., C.B., I.W., J.M., F.Z., K.K., J.S., A.B. and J.C.
contributed to review and editing of the manuscript and approved the
final manuscript.

## Competing interests

K.K. is a member of the Ethnicity Subgroup of the UK Scientific Advisory
Group for Emergencies (SAGE) and Member of SAGE. The remaining
authors declare no competing interests.

## Additional information

**Supplementary information** The online version contains
supplementary material available at

Vahé Nafilyan or Charlotte R. Bermingham.

**Peer review information** *Nature Communications* thanks Paddy Far-
rington, Anders Husby and the other, anonymous, reviewer(s) for their
contribution to the peer review of this work. Peer reviewer reports are
available.

