## [Peer Review File · Nature Communications]

Risk of death following COVID-19 vaccination or positive SARS-CoV-2 test in young people in EnglandREVIEWER COMMENTS

Reviewer #1 (Remarks to the Author):

I will thank V. Nafilyan and colleagues for writing this important manuscript. The results are interesting and the article is well-written. The manuscript is especially interesting as it is able (with large statistical power) to investigate death in young people, which is a very rare event in general and specifically following COVID-19 vaccination.

However, there are two major aspects of the manuscript I think needs to be revised/explained:

1) The two main vaccine types used in the UK (Vector(AZ) vs mRNA (Pfizer+Moderna)) have very profound differences in type of severe adverse events (VITT versus myocarditis, respectively), why the results need to be stratified. As a bare minimum it needs to be explained how large a proportion of young individuals in England were vaccinated with either type. Nevertheless, vaccine-type-stratified results would make the article much more relevant.

2) Use of SCCD for analysis of death.

I order to understand the analysis, I discussed the analytic approach with a bright statistician from my department (Emilia Myrup Thiesson). Her comments are as follows:

"Technically, this is a possible way to analyze death. Since it goes against the two assumptions in a SCCS model (no event dependent exposures and event dependent observation periods), they use an extension of the SCCS model that takes event dependent exposures into account and they extend the observation period to the end of the study instead of ending it at death.

However, that being said, I don't think this analysis is appropriate unless there is a sensitivity analysis backing it up. In the article referenced they write:

"As with all statistical models, departure for assumptions may lead to incorrect inferences, and due consideration must be given to this possibility. To this end, sensitivity studies should be undertaken, which may include some of those suggested in the present paper."

Statistically speaking, it seems more obvious to perform a survival analysis instead of using SCCS since we are, in fact, looking at survival. I would recommend having that as a sort of sensitivity analysis to back up that the assumptions made using this model hold true."

Therefore, I strongly suggest that the authors conduct an appropriate sensitivity analysis to investigate whether their assumptions hold or argue why this is not needed.

In my opinion, the manuscript is worthy of publication if the authors satisfactorily can answer these two questions.

Minor comments:

- Analysis of death after infection is primarily a reflection of pre-Omicron era mortality. Should be noted.
- Did the authors consider adjusting for multiple testing?
- Results from line 244 to 248 should go in abstract if possible
- Could the authors consider whether using positive SARS-CoV-2 test as an exposure was biased, because testing was more profound among more ill-individuals (e.g., all younger individuals admitted to a hospital)

Thank you for writing this interesting manuscript.

Kindest regards,
Anders Husby

Reviewer #2 (Remarks to the Author):

General comments

This paper describes a SCCS (self-controlled case series) investigation of the potential association

between deaths, Covid-19 infection, and Covid vaccination in persons aged 12 – 29 years in England.

The data include deaths occurring between December 2020 and March or May 2022 as recorded by ONS and HES; these data were linked by NHS number to Covid tests and vaccination records: the comprehensive data used in this study are one of its strengths. The SCCS analysis strategy is based on recently-developed methodology for analysing deaths in this context; these methods have been adapted by the authors (further clarification of these adaptations is needed, as set out below). The data are analysed with relevant sensitivity and subgroup analyses, and useful graphics in the supplementary material. The main results are presented clearly and succinctly, and discussed with due regard to study limitations.

My overall assessment, subject to some further clarification, is that the authors have undertaken a thorough analysis of their data, and produced a useful and compelling set of results.

Points for clarification

There are two points on which I felt some clarification was needed.

My main query relates to the method of analysis. The SCCS method referenced in [23, 25, 26] seems to have been adapted in two ways: first, to take account of the fact that the analysis was undertaken in weekly time units (rather than days), with an adjustment for 'incomplete' weeks; second, by inclusion of a spline term to model the temporal trend (rather than a step function). Further detail on both these modifications would be welcome, perhaps in the supplementary material.

The reason I'm asking for this is as follows. This SCCS model is not a simple loglinear one that can easily be fitted in a single step in standard software: it relies on a system of unbiased estimating equations for the parameters. (This is the case for multi-dose vaccines; for vaccine dose 3 and first covid infections, a simpler analysis would be possible.) The pseudo-likelihood used in the iterative fits is constructed specifically to generate the appropriate estimating equations, and thus to give the correct parameter estimates (confidence intervals are obtained separately, as described in [25]). It may not necessarily generalise straightforwardly (and if it so happens that it does, this needs to be demonstrated). The modifications to this procedure could therefore be non-trivial, which is why I've asked for some clarification.

Note that an analysis without these added bells and whistles would be straightforward using function `eventdependexp` within the R package SCCS associated with [25] – treating all weeks as complete and with a step function rather than splines.

Also, am I right in thinking that the hospital data are daily rather than weekly? If so, would it not be better to analyse them on a daily rather than weekly scale? This would circumvent these issues completely. Perhaps the authors might consider these simpler analyses.

My second query relates to what has come to be called the 'day 0 effect' in the analysis of Covid infection. For analyses of complications of Covid based on hospital records, and perhaps other types of data, the relative incidence estimates for the day of the positive Covid test ('day 0') has been shown to be biased upwards, often considerably so, owing to inverse causality issues associated with the testing regime on admission to hospital (see Fonseca-Rodriguez et al, *Stat Med* 40: 6197; 2021). Is this an issue here? This could be checked with the HES data, for example. If it is, might it affect the week 1 estimates after Covid infection? (At Lines 235-236 the Covid effect is described as "most pronounced" in week 1.) Day 0 estimates may also be biased for vaccine exposures – but in the other direction, for different reasons.

Minor points

Line 145: "in general, age and comorbidity...": better to replace age by age group, since age is time-varying.

Line 149: censored at death (typo).

Reference [26]: this reference does not appear in the text, I think.

Line 153: add "or uninfected" at the end?

Lines 201-2: the interpretation of the decline seems to imply a positive association with deaths before 12 weeks after dose 1, implying an RI > 1 for dose 1, yet none is observed. This might suggest that the decline over time is actually attributable to temporal effects, rather than dose 2.

Lines 288-292. The reasoning in this paragraph, relating to the well vaccinee effect, is fine; but perhaps there is also an alternative explanation: if Covid causes deaths in this population (which seems to be the case), and vaccination protects against Covid for a time, then one might expect to find an RI < 1 for deaths in the immediate post-vaccination period due to the effectiveness of the vaccine, since it later wanes.

Paddy Farrington

Reviewer #3 (Remarks to the Author):

This is a very interesting study which uses the England Hospital Episodes Statistics (HES) database to explore the relative incidence of cardiac and all-cause death after positive CoV-2-SARS and COVID-19 vaccination. The noteworthy findings are that no increased incidence rate ratio (IRR) in all-cause death, or hospital death after vaccination was found. A small increase was found for cardiac death amongst 0-29 year old females. After CoV-2-SARS infection an increased IRR was found for cardiac and all-cause death amongst the unvaccinated. For the vaccinated increased all-cause death and hospital death was found for post CoV-2-SARS infection. These data are of significance given the population coverage of HES (all 12-29 year olds who died in England) and the need to balance risks of vaccination vs. low rates of COVID-19 hospitalisations and deaths in this age group. To my knowledge there are no studies with sufficient power to investigate mortality amongst the younger age groups.

Abstract

Line 24 The size of the study population should be included.

Line 30 Comparison group is defined here as 'subsequently' - should this be the pre-vaccination & 12 week post-exposure periods?

Line 30: The sub-group analysis finding of increased cardiac death amongst vaccinated females warrants contextualisation in the abstract: e.g. For females - 1 additional cardiac registered death for every 709,826 (95% CI 466,391, 6,392,519) vaccinated. For unvaccinated on date of positive SARS-CoV-2 test there was 1 additional death for every 7,459 (95% CI 6,482, 9,272) individuals; for vaccinated there was 1 additional death for every 148,760 (98,959, 2,959,910) individuals.

Line 44 Instead of subsequent periods do you mean e.g. baseline periods (i.e non-post vaccine or infection risk periods)?]

Main: Methods: It is unclear which of the vaccines available in England are being used by this age group.

Was a clearance period prior to the exposure date considered for the analysis and if not, a reason for not including a clearance period should be provided?

Two different end dates for the study were used - 25 May 2022 for death registrations and 31 March 2022 for hospital deaths - the possible limitations of having two different end dates for the study are discussed. The authors have clarified that the extended time period for registration data is due to the coroner referrals (and the time it takes to complete these). Is the date of death included in the Death Registration and therefore a consistent end date could be used? Were there

any events in HES during this time period which were not be included in Death Registration - these may be under coroner review at the time of the study analysis?

A secondary analysis described in the discussion was carried out using HES data alone. This post-hoc analysis should be described in the methods and results reported in the supplementary material.

Results: Figure 1 - there is a protective effect of during the first two weeks after vaccination. As noted this may be a result of the healthy vaccinee effect, but could suggest a protective effect of the vaccine. This has been found elsewhere: Simpson, C.R., Kerr, S., Katikireddi, S.V. et al. Second-dose ChAdOx1 and BNT162b2 COVID-19 vaccines and thrombocytopenic, thromboembolic and hemorrhagic events in Scotland. *Nat Commun* 13, 4800 (2022). This should be noted in the discussion.

Discussion: Was there any further information available on the post-vaccine cardiac deaths for 0-29 females?

Response to reviewers

We thank the reviewers for their helpful comments and suggestions. We have addressed point by point the comments below and made improvements to the manuscript as described.

Reviewer 1

1. The two main vaccine types used in the UK (Vector(AZ) vs mRNA (Pfizer+Moderna)) have very profound differences in type of severe adverse events (VITT versus myocarditis, respectively), why the results need to be stratified. As a bare minimum it needs to be explained how large a proportion of young individuals in England were vaccinated with either type. Nevertheless, vaccine-type-stratified results would make the article much more relevant.

Thank you for the suggestion, this is a useful addition to the paper.

We have added the number and percentage of vaccines by vector (mRNA or 'not mRNA or other') for the registrations and HES datasets, by dose (lines 77-78 and 85-87). We also include the percentage of people who received the first two doses who had different doses for each of these (lines 80-82 and 87-88). Supplementary Table 1 reports vaccine types by dose, and Supplementary 2 gives the number of people who received at least one dose of either mRNA or a non mRNA vaccine. There were sufficient individuals vaccinated with both mRNA and non-mRNA vectors for the first and second doses to perform a stratified analysis. In addition, Supplementary Table 3 gives the number of mRNA and non mRNA vaccine doses for our study population received before or after 7 April 2021 (when the ChAdOx1 Astra-Zeneca vaccine was withdrawn for people aged under 30), which is discussed in lines 212-220).

We have produced the analysis by vaccine vector and broken this down further by sex. The stratified analysis is described in the methods (lines 404-411) and the results are presented in Figure 2b and Supplementary Figure 5 and described in the results section (lines 125-154) and the discussion section (207-218).

2. Use of SCCD for analysis of death.

I order to understand the analysis, I discussed the analytic approach with a bright statistician from my department (Emilia Myrup Thiesson). Her comments are as follows:

"Technically, this is a possible way to analyze death. Since it goes against the two assumptions in a SCCS model (no event dependent exposures and event dependent observation periods), they use an extension of the SCCS model that takes event dependent exposures into account and they extend the observation period to the end of the study instead of ending it at death.

However, that being said, I don't think this analysis is appropriate unless there is a sensitivity analysis backing it up. In the article referenced they write:

"As with all statistical models, departure for assumptions may lead to incorrect inferences, and due consideration must be given to this possibility. To this end, sensitivity studies should be undertaken, which may include some of those suggested in the present paper."

Statistically speaking, it seems more obvious to perform a survival analysis instead of using SCCS since we are, in fact, looking at survival. I would recommend having that as a sort of sensitivity analysis to back up that the assumptions made using this model hold true."

Therefore, I strongly suggest that the authors conduct an appropriate sensitivity analysis to investigate whether their assumptions hold or argue why this is not needed.

We acknowledge that further sensitivity tests would strengthen the analysis by testing the assumptions of the model used.

However, we do not believe that survival analysis would be an appropriate technique to investigate this research question with these data. Young people with severe health conditions were prioritised for vaccination in the UK, and we would not be able to adjust appropriately for this with the variables available. The main strength of the SCCS methodology is that it deals with this between-person confounding [1].

Ghebremichael-Weldeselassie et al propose a sensitivity analysis assessing robustness to the occurrence of deaths not due to the event in the context of the event of interest being haemorrhagic stroke, because haemorrhagic stroke may precipitate death but cannot occur after death [2]. In our case, where the event of interest is death itself, this would not be appropriate; censoring any time after death would differentially reduce the denominator time for the reference period and introduce a negative bias on effect estimates. The main potential sensitivities in our analysis are with respect to appropriate specification of the risk period, and the handling of time trends (including increasing censoring due to coroner referrals). We assessed sensitivity to specification of the risk period by exploring the full range of risk periods that could be specified within the constraints of the data, given timing of vaccine doses (Supplementary Figures 7 and 8).

We addressed time trends by accounting for calendar time in our analyses. We have also included further sensitivity analyses where calendar time is included stepwise rather than as a restricted cubic spline (Supplementary Figures 9 and 10). We have added details on this to the methods section (lines 421-423) the results section (lines 226-228).

[1] H. J. Whitaker et al, "Tutorial in biostatistics: the self-controlled case series method," *Statistics in Medicine*, vol. 25, no. 10, pp. 1768-1797, 5 2006.

[2] Y. Ghebremichael-Weldeselassie et. al., "A modified self-controlled case series method for event-dependent exposures and high event-related mortality, with application to COVID-19 vaccine safety", *Statistics in Medicine*, vol. 41, pp. 1735-1750, 2022.

3. Analysis of death after infection is primarily a reflection of pre-Omicron era mortality. Should be noted.

We have noted this in the Methods section in lines 349-351, with the dates of the pre-delta-, delta- and omicron-dominant periods as defined in this article, which we have referenced.

4. Did the authors consider adjusting for multiple testing?

We thank the reviewer for asking this pertinent question. We did not initially plan to adjust for multiple testing. Because the number of deaths in this age group are (fortunately!) rather low, there is a risk that adjusting for multiple testing would lead to mask any signal.

Following your question, we have calculated corrected p-values, using the Benjamini-Hochberg Procedure (The Benjamini-Hochberg Procedure was chosen as it was the least conservative), to the data for the breakdowns in Figure 2. Using the corrected p-values, the increase in RI for females, for

cardiac deaths after dose 1 is not significant when the Benjamini-Hochberg adjustment is applied. However, the Benjamini-Hochberg corrected p value for the non mRNA vaccine vector for all doses and first dose in females for cardiac death does produce a statistically significant result.

We have included a description of the results broken down by vaccine vector and sex in the manuscript (see response to Comment 1), highlighting the results for females for cardiac deaths after dose 1 of a non mRNA vaccine, which is included in the abstract (line 65). The relevant p values are included in Table 2 in an appendix to our responses. However, we believe that adding multiple testing adjustment at that stage could be interpreted as a way to mask a potential signal.

5. Results from line 244 to 248 should go in abstract if possible

We have had to reformat the abstract to the limited 150 words in line with Nature Communications style so unfortunately this is not feasible.

6. Could the authors consider whether using positive SARS-CoV-2 test as an exposure was biased, because testing was more profound among more ill-individuals (e.g., all younger individuals admitted to a hospital)

We thank the reviewer for this helpful suggestion. We undertook a sensitivity analysis to determine whether this was a source of bias in our data for vaccinations and infections in the HES dataset. We excluded those individuals whose positive test result for SARS-CoV-2 infection was on the same day as the start of the hospital episode after which they died. This is described in the methods (lines 348-349), the results (lines 195-201) and the discussion (lines 276-281). We did not find evidence of bias in the results for death after vaccination. We did find evidence of bias in the results for death after infection and have discussed this in the limitations section. As a result of finding this bias, we have excluded people with a positive test on admission to hospital from the HES dataset for the main analysis of infections, and included the original results where these people are included in Supplementary Figures 11 and 12.

Reviewer 2

7. My main query relates to the method of analysis. The SCCS method referenced in [23, 25, 26] seems to have been adapted in two ways: first, to take account of the fact that the analysis was undertaken in weekly time units (rather than days), with an adjustment for 'incomplete' weeks; second, by inclusion of a spline term to model the temporal trend (rather than a step function). Further detail on both these modifications would be welcome, perhaps in the supplementary material.

The reason I'm asking for this is as follows. This SCCS model is not a simple loglinear one that can easily be fitted in a single step in standard software: it relies on a system of unbiased estimating equations for the parameters. (This is the case for multi-dose vaccines; for vaccine dose 3 and first covid infections, a simpler analysis would be possible.) The pseudo-likelihood used in the iterative fits is constructed specifically to generate the appropriate estimating equations, and thus to give the correct parameter estimates (confidence intervals are obtained separately, as described in [25]). It may not necessarily generalise straightforwardly (and if it so happens that it does, this needs to be demonstrated). The modifications to this procedure could therefore be non-trivial, which is why I've asked for some clarification.

Note that an analysis without these added bells and whistles would be straightforward using function `eventdependexp` within the R package `SCCS` associated with [25] – treating all weeks as complete and with a step function rather than splines.

Also, am I right in thinking that the hospital data are daily rather than weekly? If so, would it not be better to analyse them on a daily rather than weekly scale? This would circumvent these issues completely. Perhaps the authors might consider these simpler analyses.

Thank you for this detailed comment. We followed the method referenced in [1-3] to construct our estimates. We did not use the `SCCS` package itself due to restrictions on the secure analysis environment we use but we followed the same logic. The modifications are: adjustment of calendar time using a restricted cubic spline rather than fortnightly intervals and adjustment for incomplete weeks. We have therefore conducted further sensitivity tests and simulations to confirm the validity of these adjustments.

- 1. We have added a sensitivity test where the calendar time adjustment is included as fortnightly steps, rather than a spline, obtaining consistent results with the main analysis. This is described in the methods (lines 413-414) and results (lines 191-193).*
- 2. While we do have the day of death in both the registrations and hospital datasets, we analysed the data in weeks, with adjustment for incomplete weeks, due to the impractical size of a daily dataset for model fitting. However, we have added a sensitivity test where we conducted the main vaccinations analysis (Figure 1 in the manuscript), for risk periods, using daily data and where calendar time is adjusted daily. The estimates are consistent with those of the main model, indicating that the use of weekly data with an offset produces consistent estimates to using the daily dataset. The estimates are compared in Table 1 in an appendix to this manuscript.*
- 3. We have recalculated the confidence intervals using bootstrapping as recommended in [3] for the main model (presented in Figure 1 in an appendix to our responses). These were similar to the confidence intervals produced by the model fitting, therefore we have retained the confidence intervals from the model fitting throughout the manuscript. We mention this in the method section (line 395).*

[1] Y. Ghebremichael-Weldeslassie et. al., “A modified self-controlled case series method for event-dependent exposures and high event-related mortality, with application to COVID-19 vaccine safety”, *Statistics in Medicine*, vol. 41, pp. 1735-1750, 2022.

[2] P. Farrington et al, *Self-controlled Case Series studies: A modelling Guide with R*, Boca Raton: Chapman & Hall/CRC Press, 2018.

[3] C. P. Farrington et al, “Case series analysis for censored, perturbed, or curtailed post-event exposures,” *Biostatistics*, vol. 10, no. 1, pp. 3-16, 2009.

- 8. My second query relates to what has come to be called the ‘day 0 effect’ in the analysis of Covid infection. For analyses of complications of Covid based on hospital records, and perhaps other types of data, the relative incidence estimates for the day of the positive Covid test (‘day 0’) has been shown to be biased upwards, often considerably so, owing to*

inverse causality issues associated with the testing regime on admission to hospital (see Fonseca-Rodriguez et al, Stat Med 40: 6197; 2021). Is this an issue here? This could be checked with the HES data, for example. If it is, might it affect the week 1 estimates after Covid infection? (At Lines 235-236 the Covid effect is described as “most pronounced” in week 1.) Day 0 estimates may also be biased for vaccine exposures – but in the other direction, for different reasons.

We acknowledge that there may be bias introduced due to increased testing of people who are hospitalised, which will particularly affect the HES dataset results for infections. We ran a sensitivity test for vaccinations and infections in the HES dataset: we excluded those individuals whose positive test result for SARS-CoV-2 infection was on the same day as the start of the hospital episode after which they died. This is described in the methods (lines 414-417), the results (lines 193-201) and the discussion (lines 276-281). We did not find evidence of bias in the results for death after vaccination. We did find evidence of bias in the results for death after infection and have discussed this in the limitations section. As a result of finding this bias, we have excluded people with a positive test on admission to hospital from the HES dataset for the main analysis of infections, and included the original results where these people are included in Supplementary Figure 11.

9. Line 145: “in general, age and comorbidity...”: better to replace age by age group, since age is time-varying.

We have replaced age with ‘age group’ (line 373)

10. Line 149: censored at death (typo).

We have corrected this (line 377)

11. Reference [26]: this reference does not appear in the text, I think.

We have added a citation for this reference (line 387), now reference [35].

12. Line 153: add “or uninfected” at the end?

We have added in this omission (lines 379-382)

13. Lines 201-2: the interpretation of the decline seems to imply a positive association with deaths before 12 weeks after dose 1, implying an RI > 1 for dose 1, yet none is observed. This might suggest that the decline over time is actually attributable to temporal effects, rather than dose 2.

We agree that a decline in deaths after dose 1 could be temporal as well as due to dose 2. However, it is unlikely in this case to be temporal as first vaccinations were given out over a range of calendar times so a drop would be unlikely to be sudden and the drop after dose 3, where there would not be a dose effect, is much more gradual indicating that a temporal drop after dose 1 would also be more gradual than is observed.

Temporal effects such as registration delay that means deaths are less likely to be observed at later calendar times are taken into account in our model by the inclusion of calendar time. Dose effects are taken into account by the structure of the stacked dataset, which is designed to deal with event-dependent exposures and multiple doses of vaccination [1]. Therefore, a change in RI after vaccination should not be related to either temporal effects or dose effects.

[1] Y. Ghebremichael-Weldeselassie et. al., "A modified self-controlled case series method for event-dependent exposures and high event-related mortality, with application to COVID-19 vaccine safety", *Statistics in Medicine*, vol. 41, pp. 1735-1750, 2022.

14. Lines 288-292. The reasoning in this paragraph, relating to the well vaccinee effect, is fine; but perhaps there is also an alternative explanation: if Covid causes deaths in this population (which seems to be the case), and vaccination protects against Covid for a time, then one might expect to find an $RI < 1$ for deaths in the immediate post-vaccination period due to the effectiveness of the vaccine, since it later wanes.

In our results the protective effect is only seen in the first two weeks after vaccination. A decrease in protection after the first 2 weeks could produce these results.

However, this is very unlikely to occur; vaccine effectiveness is lower in the first few weeks after vaccination as antibodies increase to threshold level [1], there is a lead time of 2-4 weeks between positive test and death [2], and the timescale for waning to occur is much longer than a few weeks and barely affects younger age groups [3].

[1] J. Lopez Bernal et. al., "Effectiveness of the Pfizer-BioNTech and Oxford-AstraZeneca Vaccines on covid-19 related symptoms, hospital admissions and mortality in older adults in England: test negative case-control study", *The BMJ*, vol. 373, p. n1088, 2021.

[2] T. Ward et. al., "Understanding an evolving pandemic: An analysis of the clinical time delay distributions of COVID-19 in the United Kingdom", *PLoS ONE*, vol. 16, no. 10, p. e0257978, 2021.

[3] C. Menni et. al., "COVID-19 vaccine waning and effectiveness and side-effects of boosters: a prospective community study from the ZOE COVID Study", *The Lancet Infectious Diseases*, vol. 22, pp 1002-1010, 2022.

Reviewer 3

15. Line 24 The size of the study population should be included.

We have had to reformat the abstract to be 150 words in line with the format of a Nature Communications paper, therefore cannot include this information in the abstract. The size of the study population is included in the results section (lines 74 and 84).

16. Line 30 Comparison group is defined here as 'subsequently' - should this be the pre-vaccination & 12 week post-exposure periods?

The risk period is the 12 weeks after each vaccination dose/infection. The reference period is all other weeks post vaccination/infection. The pre-vaccination/infection weeks do not form part of the reference group as no events (deaths) can occur prior to the exposure. The pre-vaccination/infection weeks are included in order to adjust for time-dependent variation, along with unvaccinated/uninfected individuals.

We have clarified this in the manuscript, changing 'subsequently' to 'more than 12 weeks after any dose' on lines 29-30 and 62-63 in the revised abstract and also updated the labels in Tables 1 and 2.

17. Line 30: The sub-group analysis finding of increased cardiac death amongst vaccinated females warrants contextualisation in the abstract: e.g. For females - 1 additional cardiac registered death for every 709,826 (95% CI 466,391, 6,392,519) vaccinated. For unvaccinated on date of positive SARS-CoV-2 test there was 1 additional death for every 7,459 (95% CI 6,482, 9,272) individuals; for vaccinated there was 1 additional death for every 148,760 (98,959, 2,959,910) individuals.

We have had to reformat the abstract to be 150 words in line with the format of a Nature Communications paper, therefore cannot include this information in the abstract.

18. Line 44 Instead of subsequent periods do you mean e.g. baseline periods (i.e non-post vaccine or infection risk periods)?

We have clarified this in the manuscript, changing 'subsequently' to 'more than 12 weeks after any dose' on lines 29-30 and 62-63 in the revised abstract and also updated the labels in Tables 1 and 2.

19. Main: Methods: It is unclear which of the vaccines available in England are being used by this age group.

We have added the percentage of vaccines by vector (mRNA or 'not mRNA or other') for the registrations and HES datasets, by dose (lines 77-80 and 86-87 and Supplementary Table 1). We have also added the number and percentage of people with two doses who had a different vector for first and second dose (lines 81-82 and 88-89 and Supplementary Table 2). In addition, Supplementary Table 3 gives the number of mRNA and non mRNA vaccine doses for our study population received before or after 7 April 2021 (when the ChAdOx1 Astra-Zeneca vaccine was withdrawn for people aged under 30), which is discussed in lines 212-220).

20. Was a clearance period prior to the exposure date considered for the analysis and if not, a reason for not including a clearance period should be provided?

Inclusion of clearance period is appropriate in conventional SCCS designs where there the event of interest only delays subsequent vaccination. However, when the event of interest is death then this cannot occur prior to vaccination and, in the SCCS design employed to address this issue, time before vaccination does not contribute directly to estimation of the exposure effect.

21. Two different end dates for the study were used - 25 May 2022 for death registrations and 31 March 2022 for hospital deaths - the possible limitations of having two different end dates for the study are discussed. The authors have clarified that the extended time period for registration data is due to the coroner referrals (and the time it takes to completed these). Is the date of death included in the Death Registration and therefore a consistent end date could be used? Were there any events in HES during this time period which were not be included in Death Registration - these may be under coroner review at the time of the study analysis?

Thank you for the suggestion. We have added to the text (lines 324-325) the count of deaths in HES that are not in death registrations.

The date of death is recorded on the death registrations. Due to the low numbers of cases, we used all available data at the time for each analysis. This includes death registrations that occurred up to 25 May 2022 and were recorded by 8 June 2022, rather than limiting the data at the maximum recorded date of death in the HES dataset (31 March 2022). The analysis of SARS-CoV-2 positive tests

is limited to 31 December 2022 for both HES and registrations datasets, as this was the latest Test and Trace data available to us.

22. A secondary analysis described in the discussion was carried out using HES data alone. This post-hoc analysis should be described in the methods and results reported in the supplementary material.

Thank you for identifying this. We had mistakenly referred to the HES data analysis as a secondary analysis in the discussion when this analysis is in fact part of the main analysis and presented alongside the death registrations analysis. It is therefore already described in the methods and reported in the main results section. We have clarified this in the discussion (lines 270-271).

23. Results: Figure 1 - there is a protective effect of during the first two weeks after vaccination. As noted this may be a result of the healthy vaccinee effect, but could suggest a protective effect of the vaccine. This has been found elsewhere: Simpson, C.R., Kerr, S., Katikireddi, S.V. et al. Second-dose ChAdOx1 and BNT162b2 COVID-19 vaccines and thrombocytopenic, thromboembolic and hemorrhagic events in Scotland. *Nat Commun* 13, 4800 (2022). This should be noted in the discussion.

A protective effect that decreases after only 2 weeks after vaccination is unlikely to occur; vaccine effectiveness is lower in the first few weeks after vaccination as antibodies increase to threshold level [1], there is a lead time of 2-4 weeks between positive test and death [2], and the timescale for waning to occur is much longer than a few weeks and barely affects younger age groups [3].

*[1] J. Lopez Bernal et. al., "Effectiveness of the Pfizer-BioNTech and Oxford-AstraZeneca Vaccines on covid-19 related symptoms, hospital admissions and mortality in older adults in England: test negative case-control study", *The BMJ*, vol. 373, p. n1088, 2021.*

*[2] T. Ward et. al., "Understanding an evolving pandemic: An analysis of the clinical time delay distributions of COVID-19 in the United Kingdom", *PLoS ONE*, vol. 16, no. 10, p. e0257978, 2021.*

*[3] C. Menni et. al., "COVID-19 vaccine waning and effectiveness and side-effects of boosters: a prospective community study from the ZOE COVID Study", *The Lancet Infectious Diseases*, vol. 22, pp 1002-1010, 2022.*

24. Discussion: Was there any further information available on the post-vaccine cardiac deaths for 0-29 females?

We include in our original manuscript the number of females aged 12-29 that would need to be vaccinated for 1 additional cardiac registered death to occur (line 134) to add more context to the increased relative incidence found in this subgroup. We also breakdown the results by vaccine vector and sex. The stratified analysis is described in the methods (lines 404-411) and the results are presented in Figure 2b and Supplementary Figure 5 and described in the results section (lines 125-154) and the discussion section (207-218). These results show that the post-vaccine cardiac deaths in females are driven by the first dose of a non mRNA vaccine. We include the absolute risk for cardiac deaths non mRNA vaccines for females in the abstract (line 31-32).

In published literature, an increased risk of myocarditis has generally been found, for people vaccinated with an mRNA vaccine, to be higher among males than females [1, 2, 3]. However, one study found an increased risk among women [4].

References 1, 2 and 4 are cited in the discussion (lines 241-242). We have added to this sentence to mention that a greater increased risk is generally found among young males and included reference 3, which is a review of literature on this topic. Additionally, we have included a paragraph in the discussion describing a recently published study in Florida that found an elevated risk of cardiac death in the first four weeks after mRNA vaccination in people ages 18-39 years (lines 221-237).

[1] M. E. Oster et. al., "Myocarditis cases reported after mRNA-based COVID_19 vaccination in the US from December 2020 to August 2021", JAMA, vol. 327, no. 4, pp. 331-340, 2022.

[2] M. Patone et. Al., "Risk of myocarditis following sequential COVID-19 vaccinations by age and sex", medRxiv, 2021.

[3] J. Pillay et. Al, "Incidence, risk factors, natural history, and hypothesised mechanisms of myocarditis and pericarditis following covid-19 vaccination: living evidence syntheses and review", The BMJ, vol. 378, p. e069445, 2022.

[4] A. Husby et. Al, "SARS-CoV-2 vaccination and myocarditis or myopericarditis: population based cohort study", The BMJ, vol. 375, pp 1-9, 2021.

Appendix

Figure 1 | Estimates for relative incidence of all-cause registered death in the risk period, weeks 1-12 after COVID-19 vaccination, compared to the baseline period. Death registrations.

Table 1 | Relative incidence of cardiac death and all-cause death in the risk period after vaccination as a whole (1-12 weeks), compared to the baseline period for the weekly data (Figure 1 in the manuscript) and daily data, where calendar time is adjusted by day rather than week.

Cause	Dose	Incidence rate ratio from weekly data (main results)	Incidence rate ratio from daily data
All-cause registered deaths	All doses	0.88 (0.80, 0.97)	0.88 (0.80, 0.97)
	Dose 1	0.95 (0.84, 1.07)	0.95 (0.84, 1.07)
	Dose 2	0.79 (0.68, 0.91)	0.79 (0.68, 0.91)
	Dose 3	0.71 (0.55, 0.93)	0.71 (0.55, 0.93)
Cardiac registered deaths	All doses	1.11 (0.87, 1.42)	1.11 (0.87, 1.42)
	Dose 1	1.20 (0.85, 1.68)	1.19 (0.85, 1.68)
	Dose 2	1.01 (0.69, 1.47)	1.01 (0.69, 1.48)
	Dose 3	0.98 (0.50, 1.92)	0.99 (0.51, 1.92)
Hospital registered deaths	All doses	0.89 (0.77, 1.04)	0.89 (0.77, 1.04)
	Dose 1	0.96 (0.78, 1.18)	0.96 (0.78, 1.17)
	Dose 2	0.80 (0.63, 1.02)	0.80 (0.63, 1.02)
	Dose 3	0.86 (0.55, 1.33)	0.86 (0.55, 1.32)

Table 2 | Raw and Benjamini-Hochberg corrected p-values for the results where there is an increased risk of cardiac or all-cause death, for the results for deaths following COVID-19 vaccination broken down by sex, and by both sex and vaccine vector.

Value	IRR	p-value (uncorrected)	Benjamini-Hochberg corrected p-value
Female, first dose, cardiac	1.79 (1.05, 3.05)	0.03	0.27
Female, first dose, non mRNA vaccine vector, all-cause	1.66 (1.10, 2.51)	0.01	0.07
Female, first dose, non mRNA vaccine vector, cardiac	3.52, (1.71, 7.25)	<0.001	0.006
Female, all doses combined, non mRNA vaccine, all-cause	1.55 (1.10, 2.18)	0.01	0.07
Female, all doses combined, non mRNA vaccine, cardiac	3.02 (1.65, 5.53)	<0.001	0.006

REVIEWERS' COMMENTS

Reviewer #1 (Remarks to the Author):

The authors have done a wonderful job answering the review questions. This has now become a very important article, that definitively deserves publication.

I only have to minor comments regarding the Discussion:

- when referencing the 'Florida' data, please underline that it is not peer-reviewed and not published in a scholarly journal, why I would at most define it as a 'pre-print' or simply 'data analysis' (and not a study yet). However, the discussion of this data is important, why it should be kept in the manuscript.

- I would delete the section on the "preparatory simulation study" unless it is reported. Instead, publish or pre-print this study fast for reference.

Again, this is great work and I thank the authors for their efforts.

Anders Husby/reviewer 1

Reviewer #2 (Remarks to the Author):

The authors have improved their interesting paper in several ways. In my view they have responded adequately to the comments I made in my original review, pertaining notably to the adaptation of the SCCS method and the "day zero" issue. In particular, they have undertaken additional analyses that help to validate their approach.

I have no further comments. I recommend publication.

Response to reviewers: Risk of death following COVID-19 vaccination or positive SARS-CoV-2 test in young people in England

We thank the reviewers for their valuable comments and suggestions during the review process and their positive response to this revised manuscript.

We have addressed the following comments on the revised manuscript from Reviewer 1.

“When referencing the 'Florida' data, please underline that it is not peer-reviewed and not published in a scholarly journal, why I would at most define it is a 'pre-print' or simply 'data analysis' (and not a study yet). However, the discussion of this data is important, why it should be kept in the manuscript.”

We thank the reviewer for the suggestion and agree that this distinction is important. We have edited the first sentence of the second paragraph of the discussion, which previously read:

“A recent **study based in Florida** (United States) found increased risk of cardiac death in the first four weeks after mRNA vaccination in people aged 18–39 years...”

to read:

“A recent **analysis of data from Florida** (United States) found increased risk of cardiac death in the first four weeks after mRNA vaccination in people aged 18–39 years...”

“I would delete the section on the "preparatory simulation study" unless it is reported. Instead, publish or pre-print this study fast for reference.”

Thank you for the suggestion. We have removed reference to the preparatory simulation study and edited the second sentence of the second last paragraph of the discussion, which previously read:

“A preparatory simulation study (not reported) indicated negative bias with risk periods that exceeded dosing intervals, which is consistent with the negative effects observed in sensitivity analysis with risk periods of more than twelve weeks “

To read:

“The negative effects observed in sensitivity analysis with risk periods of more than twelve weeks suggest a negative bias with risk periods that exceeded dosing intervals.”